# Validation and Optimization of Proximal Femurs Microstructure Analysis Using High Field and Ultra-High Field MRI

**DOI:** 10.3390/diagnostics11091603

**Published:** 2021-09-02

**Authors:** Enrico Soldati, Jerome Vicente, Daphne Guenoun, David Bendahan, Martine Pithioux

**Affiliations:** 1Aix Marseille Univ, CNRS, IUSTI, 13453 Marseille, France; jerome.vicente@univ-amu.fr; 2Aix Marseille Univ, CNRS, CRMBM, 13385 Marseille, France; david.bendahan@univ-amu.fr; 3Aix Marseille Univ, CNRS, ISM, 13288 Marseille, France; daphne.guenoun@ap-hm.fr (D.G.); martine.pithioux@univ-amu.fr (M.P.); 4Department of Radiology, Institute for Locomotion, Sainte-Marguerite Hospital, Aix Marseille Univ, APHM, CNRS, ISM, 13274 Marseille, France; 5Department of Orthopaedics and Traumatology, Institute for Locomotion, Sainte-Marguerite Hospital, Aix Marseille Univ, APHM, CNRS, ISM, 13274 Marseille, France

**Keywords:** osteoporosis, ultra-high field MRI, μCT, turbo spin echo, cadaveric human femur, bone morphology, air bubbles artefacts, gradient echo, bone microarchitecture, biomechanical fracture test

## Abstract

Trabecular bone could be assessed non-invasively using MRI. However, MRI does not yet provide resolutions lower than trabecular thickness and a comparative analysis between different MRI sequences at different field strengths and X-ray microtomography (μCT) is still missing. In this study, we compared bone microstructure parameters and bone mineral density (BMD) computed using various MRI approaches, i.e., turbo spin echo (TSE) and gradient recalled echo (GRE) images used at different magnetic fields, i.e., 7T and 3T. The corresponding parameters computed from μCT images and BMD derived from dual-energy X-ray absorptiometry (DXA) were used as the ground truth. The correlation between morphological parameters, BMD and fracture load assessed by mechanical compression tests was evaluated. Histomorphometric parameters showed a good agreement between 7T TSE and μCT, with 8% error for trabecular thickness with no significative statistical difference and a good intraclass correlation coefficient (ICC > 0.5) for all the extrapolated parameters. No correlation was found between DXA-BMD and all morphological parameters, except for trabecular interconnectivity (R^2^ > 0.69). Good correlation (*p-*value < 0.05) was found between failure load and trabecular interconnectivity (R^2^ > 0.79). These results suggest that MRI could be of interest for bone microstructure assessment. Moreover, the combination of morphological parameters and BMD could provide a more comprehensive view of bone quality.

## 1. Introduction

Low bone mineral density (BMD) and microarchitectural alterations are both responsible for osteoporosis, a bone disorder that leads to an increased sensitivity to fractures [1]. The economic burden of osteoporosis in Europe was estimated to be 37 billion euros (28 million patients) in 2010 and is expected to increase by 25% by2025 [2]. To alleviate this economic pressure and increase the quality of life of patients, the early diagnosis of osteoporosis is a critical issue. Osteoporosis is currently diagnosed on the basis of bone mineral density (BMD) measurements using dual-energy X-ray absorptiometry (DXA). Previous studies have shown that DXA has poor sensitivity. Schuil et al. reported that only 44% of all non-vertebral fractures occurred in women with a T-score lower than −2.5 SD, and in men, this percentage is even lower (21%) [3]. Although the DXA sensitivity increases to 88% for both men and post-menopausal women when considering DXA-determined osteoporosis and low bone density (T-score below −2.0 SD), the ability to discriminate the healthy patients, i.e., specificity, is poor (around 41% for post-menopausal woman and 55% for men) [4], resulting in a low clinical discriminative accuracy (70%) [5,6]. In addition, DXA measurements do not take into consideration microarchitectural alterations, which are also part of the structural picture of osteoporosis. Considering the whole set of aspects, the diagnostic accuracy of osteoporosis might be expected to increase. Magnetic Resonance Imaging (MRI), quantitative computed tomography (qCT), and high-resolution peripheral computed tomography (HRpQCT) can be used to assess bone microarchitecture. Whereas qCT and HRpQCT are highly radiative techniques (>100 times larger than DXA) [7,8], MRI is recognized as totally non-invasive (Figure 1).

So far, most of the MRI studies on bone microstructure have been carried out at conventional (1.5T) and high (3T) magnetic fields. Ultra-high magnetic field (7T) MRI offers a higher resolution and has been used more recently [9,10,11]. Most of the time, superficial bones such as distal radii and tibiae have been assessed [12,13,14,15], while deeper bones such as the proximal femur microarchitecture have been more scarcely analyzed [1,16]. From a technical point of view, different MRI pulse sequences such as gradient recalled echo (GRE) and spin echo (SE) have been used. Gradient-echo-type sequences have shown to provide the highest possible signal-to-noise (SNR) efficiency, while spin echo sequences are less prone to off-resonance intravoxel signal cancellation due to susceptibility differences between bone and bone marrow [11]. The corresponding results regarding bone volume fraction (BVF), trabecular thickness (Tb.Th), spacing (Tb.Sp), and number (Tb.N) have been compared to DXA [1,9], qCT [11,12,13] and X-ray microtomography (μCT) [17,18,19]. Using linear regression, good correlation coefficients (r) have been reported between 3T and 7T MRI and qCT on in vivo and ex vivo radii and tibiae (r > 0.69) [11,13], between qCT and μCT on cadaveric radii (r > 0.89) [18], and moderate to good correlation (0.53 < r < 0.87) was found between 1.5T MRI and μCT on cadaveric specimens of distal radii [17]. MRI can suffer from partial volume effects that could be responsible for the discrepancies between MRI and X-ray techniques. Previous studies have shown that spin echo (SE) sequences would provide more accurate results regarding trabecular characteristics due to the fact that they are less prone to susceptibility-induced broadening of the trabeculae [11,13]. The use of these pulse sequences might be problematic using ultra-high field (UHF) MRI considering power deposition issues, which reduces the number of acquirable images in a single scan [20]. However, the full 3D acquisition of the thickest bones could be assessed using multiple scans in a single MRI session. On that basis, the higher resolution provided by UHF MRI might be compromised if one intends to use SE sequences.

In that context, a comparative analysis between high and ultra-high field MRI of intact proximal femurs, taking into consideration the issue of pulse sequences (SE vs. GRE) would be of great interest. Such a comparative analysis could address the issues of image resolution needed for trabecular microarchitecture assessment, partial volume effects and specific power deposition of each imaging modality. For ex vivo measurements, susceptibility effects related to air bubbles trapped in the trabecular network should also be considered. Considering that these details are missing in the literature, in the present study, we intended to compare the trabecular characteristics of three intact femur heads using μCT and 7T MRI using both SE and GRE pulse sequences. We aimed at identifying the effects of different techniques (μCT vs. high-field MRI vs. UHF MRI) and pulse sequences (SE vs. GRE) on the trabecular network characteristics and to compare the corresponding metrics using those from µCT as the gold standard. Moreover, microarchitecture characteristics from all the different imaging modalities were also compared with the failure load assessed during biomechanical compression tests. The corresponding results should assess the link between trabecular characteristics and proximal femur risk of fracture and offer an assessment frame of the potential use of MRI as a non-invasive tool of bone microarchitecture in vivo.

## 2. Materials and Methods

### 2.1. Sample Preparation

All procedures were carried out in accordance with the ethical standards of the responsible committee on human experimentation of the thanatopraxy laboratory, University School of Medicine, Hôpital de la Timone, Marseille, France, that provided the bodies coming from donation and in accordance with the Helsinki Declaration of 1975, as revised in 2000.

Three cadaveric femurs (S1, S2, S3), were initially scanned using conventional DXA (total femur BMD equal to 0.83 g/cm^2^, 1.31 g/cm^2^ and 0.50 g/cm^2^, respectively for S1, S2 and S3) and then prepared according to an original vacuum procedure as previously reported [21] in order to remove air bubbles trapped in the trabecular network. After an overnight defreezing process, the femur was placed inside a plastic jar filled with a physiological solution doped with 1 mM Gd-DTPA [21]. The container was then placed on a vibrating surface while low pressure cycles where applied. Each cycle had 5 min of active pumping (pressure lower than 50 mbar) and 5 min of resting time (150 mbar). Different vibrating amplitudes (0.1 to 1.5 mm) were used to cope with different bubble size displacement (from 20 μm to 2.5 mm diameter). The application of three cycles ensured the removal of more than 99% of air bubbles [21].

### 2.2. Imaging

#### 2.2.1. µCT Imaging

μCT images of the three samples were acquired using an Rx-Solution EasyTom XL ULTRA microtomograph (Rx-Solution, Chavanod, France) [22], with a 150 kV X-ray Hamamatsu Tube, allowing a focus spot size of 5 μm. In order to acquire the complete femur head volume, an isovolumetric voxel size of 0.05 mm was chosen using an X-ray source voltage of 150 kV, a current of 343 mA, a frame rate of 8 images/s, and 1440 projections over 360 degrees of rotation. Each projection resulted from the average of 10 images in order to enhance the signal-to-noise ratio. The acquisition time was approximately 40 min.

#### 2.2.2. MRI Imaging

All the three femurs were scanned using two pulse sequences (turbo spin echo (TSE) and gradient recalled echo (GRE)) using UHF MRI (7T MAGNETOM, Siemens Healthineers, Erlangen, Germany). For comparative purposes, one femur head (S1) was also scanned at 3T (Verio Siemens Healthineers, Erlangen, Germany) with both TSE and GRE sequences. The 7T MRI was performed using a 28Ch Knee coil, while 3T MRI acquisitions were performed using a flexible 16Ch Heart coil. The corresponding sequence parameters were similar to those used in the literature [11,23,24] and adapted to our sample size (Table 1). The acquisition time was set to be acceptable for clinical applications (14 ± 4 min), while the voxel size was pushed to the machine limit.

### 2.3. Image Analysis

#### 2.3.1. Image Registration

For each of the three scanned femurs, MRI and μCT images were co-registered in the coronal plane thanks to a 3D-printed plastic registration tool positioned inside the plastic jar. The μCT alignment along the coronal plane was performed during the post-processing volume reconstruction step. For the MRI acquisitions, this alignment was performed before the acquisition. After the manual 3D alignment, an automatic 2D registration was used between one MR slice and a stack of 60 consecutive μCT slices (N_reg_) centered in the MRI absolute location to find the µCT image that better corresponded to the MRI one. The registration efficiency was assessed on the basis of the normalized cross-correlation (NCC), which measures the similarity between template and image by searching the location of the maximum value in the image matrices, as previously reported [25]. For each stack of 60 µCT images, we selected the µCT slice with the highest NCC score as the slice that better registered with the corresponding MRI one. The MRI images were first upscaled to the µCT matrix size (1785 × 1380 pixels) using the bilinear interpolation in order to work in a unique reference frame [26]. The registration was then performed using a MATLAB (MathWorks, R2020b) built-in function, imregister, with a multimodal approach and a geometric affine transformation (Figure 2). Moreover, we defined the number of μCT images between two consecutive best-registered μCT slices as ΔIm. Considering that MR images were recorded with a space between slices equal to the slice thickness and in order to reduce the cross-talk effects, we also defined the distance between consecutive MRI slices as the expected ΔIm (expΔIm) (expressed in the number of µCT images). The expΔIm, which is unique for each MRI sequence, was calculated from the ratio between the sum of slice thickness and space between slices, and the µCT slice thickness. It is now possible to evaluate the registration quality by comparing the ΔIm mean values found between two consecutive best-registered µCT slices and the expΔIm. Moreover, to maximize the region of interest (ROI), the registration process was performed on the 3 central MRI slices characterized by the higher femur head surface. Furthermore, since 3T and 7T MR images were targeting the same central bone region, we obtained very close registered μCT presenting similar morphological parameters values. Thus, we decided to only report the series of μCT data corresponding to the 7T TSE registration here. The morphological parameter analysis was performed on 2D slices.

#### 2.3.2. Bone Morphological Quantification

Conventional histomorphometric parameters were quantified and compared in registered μCT/MR images using their original resolutions. Since the binarization of the solid part of MR images was not trivial, the Sauvola filter, an automatic local thresholding technique particularly useful when the background is not uniform with a window size of 10 × 10 pixels, was applied [27] to eliminate possible biases due to manual thresholding and to take into account important contrast variations observed in images. The segmentation of the µCT images was straightforward, since the contrast was high and the voxel size was smaller than the trabecular thickness.

Three independent parameters were extrapolated from all the binarized ROIs for the multimodal and multiscale comparative analysis [7,20]. The bone volume fraction (BVF) was calculated as the ratio between bone volume and total volume. The trabecular thickness (Tb.Th) and the trabecular spacing (Tb.Sp) were extrapolated using the distance transformation map, from which the aperture map was derived using the software iMorph (iMorph_v2.0.0, AixMarseille University, Marseille, France) [28,29]. The aperture map previously used for the 2D and 3D morphology evaluation of porous matters in several fields [28] gives the diameter of the maximal disk totally enclosed and containing this voxel for every pixel. Tb.Th and Tb.Sp were then deduced from the mean values of the aperture map distribution, respectively, in the solid and the marrow phase. The trabecular number (Tb.N) was derived from the ratio between BVF and Tb.Th. The aperture map has been previously used for the 3D morphological evaluation of porous materials in different fields [19,28]. The aperture map approach, compared to the commonly used mean intercept length technique, provides local information with a sub-voxel precision [28,30].

Three additional morphological parameters were also assessed, i.e., the principal and secondary trabecular orientation (Tb.OrP and Tb.OrS, respectively) and the trabecular interconnectivity (Tb.Int). The local orientation of each pixel was computed using multiple 2D local Hessian matrices, each one resulting from the second-order derivatives of the gray level image convolved with a Gaussian matrix of fixed standard deviations (σ from 1 to 5). The eigenvalues and eigenvectors of the five 2D Hessian matrices were calculated and the eigenvectors corresponding to the largest eigenvalues were kept. The orientation was then calculated from the four-quadrant inverse tangent (tan^−1^) from the eigenvectors translating the main orientation [31]. The orientation distribution was computed considering the local orientation of binarized solid voxels only. In order to identify the first and second main directions of the trabeculae, the 2 Gaussian curves fitting method was applied using an in-house MATLAB code based on built-in function *fitnlm*, which was adapted to resolve the following model:$$Y=a+m*x+\frac{1}{\sigma_{1}\sqrt{2\pi }}*e^{-\frac{1}{2}{\left(\frac{x-\mu_{1}}{\sigma_{1}}\right)}^{2}}+\frac{1}{\sigma_{2}\sqrt{2\pi }}*e^{-\frac{1}{2}{\left(\frac{x-\mu_{2}}{\sigma_{2}}\right)}^{2}}$$
where a is the y-intercept, m is the slope, and $\mu_{1},\;\sigma_{1}$ and $\mu_{2},\;\sigma_{2}$ are, respectively, the mean and SD of the first and second Gaussian curves.

Tb.OrP was expressed as the mean ± SD of the principal fitted Gaussian curve; Tb.OrS is presented as the difference between absolute mean secondary orientation and the main principal orientation. Trabecular interconnectivity was computed as the standard deviation of the whole trabecular orientation distribution from the main principal trabecular orientation. Tb.Int represents the trabecular orientation variability and could provide information about the bone adaptability to stresses coming from different directions (Figure 3).

A one-way analysis of variance (factor = imaging modality) was carried out to assess the statistical effect of the imaging modality on the different morphological parameters. A *p*-value lower than 0.01 was considered as statistically significant. Linear regression was performed to address the functional relationship between the different imaging modalities. In addition, a Bland–Altman (BA) analysis was used to assess the agreement between the imaging modalities. The BA analysis allows the identification of any systematic difference between the measurements or possible outliers, and it is usually used to investigate any possible relationship of discrepancies between the measurements and the true value. Intraclass correlation coefficients (ICCs) were also calculated as previously reported [32] and the agreement was considered as low (ICC < 0.5), good (0.5 < ICC < 0.75) or excellent (ICC > 0.75). Moreover, the clinical relevance of bone microarchitecture was assessed by calculating the linear regression between BMD, determined from the DXA scan, and the morphological parameters computed from the image analysis of the different imaging techniques and sequences.

#### 2.3.3. Bone Mineral Density Assessment

BMD was also assessed using µCT and 7T MR images for comparison purposes. Two approaches were proposed and applied: one volumetric method, using all µCT images which provided a complete 3D bone reconstruction [33,34], and one areal method, using the µCT and MR images with higher 2D bone surface, hence corresponding to the central coronal plane.

The BMD-DXA from the specific region of the femur neck (BMD = 0.78, 0.96 and 0.54 g/cm^2^ for S1, S2 and S3, respectively) were retrieved and the ROI used to calculate it was individuated. Femur neck ROI was chosen as it represents one of the regions more affected by fragility fractures due to osteoporosis [35] and one of the most commonly occurring due to sideways fall. The angle between the long bone axis and the femur head was individuated and the stack of images were oriented to obtain the ROI of the femur neck perpendicular to the *y*-axis (see Figure 4).

Firstly, using all the µCT images, a volumetric region of interest (VOI) in correspondence of the ROI used to evaluate the femur neck BMD-DXA was individuated manually. The number of bone voxels was assessed by applying the same threshold as described in the previous section. Bone mineral content (BMC_(1)_) was calculated as the product of bone voxels for the bone density adjusted for porosity (ρ = 1.2 g Ca/cm^3^) [36]. To provide the BMD comparable to conventional BMD-DXA (expressed in g/cm^2^), in case of 3D µCT, the BMC_(1)_ was divided by the bone surface of the central coronal plane of the VOI analyzed.

The same concept was applied to calculate BMD from a 2D image for both µCT and 7T MRIs. After the positioning of the ROI in accordance with the ROI used for the DXA analysis in the femur neck, the central coronal image in the ROI was selected and binarized. The vector crossing the bone surface in the middle was individuated and used as the rotation axis. The volume that each bone pixel depicts over a rotation of 180° around the rotation axis was calculated using the volume ring formula, hence integrating the 2D surface to obtain an apparent bone volume (V_app_) of the femur neck. The calculation was performed assuming unit height, and major (R) and minor (r) radii were assessed with respect to the rotation axis (Figure 4). The rotation of the 2D surface around the central coronal plane of the femur neck was made on the basis of the femur neck anatomy. In fact, the cortical thickness, which represents the most representative bone region for the BMD, is not homogeneous. The bottom cortical layer of the femur’s neck is thicker than the upper layer. On that basis, a rotation over 2π of a surface expressing both the superior and inferior cortical layers, would provide a more reliable approximation of the total femur neck volume. Once obtained, the apparent bone volume, BMC_(2)_ and BMD were derived as previously described. Finally, to assess the correlation between techniques, the linear regression was computed between BMD derived from DXA scans and those calculated from µCT and 7T MRIs in the same femoral neck ROI.

### 2.4. Mechanical Testing

Each specimen was loaded to failure in a universal testing machine (Instron 5566, Instron, Canton, MA, USA). Each femur was placed within the loading apparatus so as to simulate a sideways fall on the greater trochanter [37,38]. Each specimen was first fixed in resin (Epoxy Axon F23) at 15° internal rotation and then the femoral heads were oriented at 10° adduction within the testing machine. The load was applied to the greater trochanter (displacement rate 10 mm/min) through a pad, which simulated a soft tissue cover, and the femoral head was covered with resin to ensure force distribution over a larger surface area. Failure load (in MPa) was defined as the first local maximum, after which the load declined by more than 10% divided by the bone surface at the fracture site [39]. Fractures were visually classified according to clinical criteria (femoral neck, intertrochanteric, subtrochanteric, or isolated greater trochanteric fractures) [9]. Finally, to assess the correlation between mechanical tests and bone morphology, the linear regression between fracture load and both BMD (DXA-derived and derived from µCT and MR images) and microarchitectural parameters was computed and the coefficient of determination (R^2^) was reported.

## 3. Results

### 3.1. Registration Quality

The optimal registration between MRIs and μCT slices are illustrated in Figure 5. In Table 2, the mean and standard deviation of the NCC scores for the femur (S1), scanned both at 3T and 7T, are presented. Similar results were obtained between 7T MR and μCT images for the other two samples. As indicated, the results showed that the registration was more efficient using the TSE sequence as compared to the GRE. The corresponding increase was 20% at 3T and 7% at 7T. Considering the GRE images, an improved registration (13%) was quantified at 7T as compared to 3T. The registration efficiency scores were similar at 3T (0.94) and 7T (0.93) regarding TSE images. Moreover, the ΔIm mean values were found to be in the same range of the expΔIm values (presented in brackets) which are the expected number of µCT slices between two consecutive MR images. More particularly, the 7T TSE showed the lowest standard deviation. In addition, considering all three samples scanned at 7T, TSE images showed generally higher efficiency than GRE images (0.91 ± 0.03 vs. 0.87 ± 0.02). Moreover, the ΔIm mean values were found in the same range to the expΔIm for both techniques (TSE and GRE) but TSE showed lower standard deviation (66 ± 9 (59) vs. 38 ± 14 (39), respectively, for TSE and GRE at 7T). Overall, the registration results showed that we were able to register multimodal and multiscale images with different voxel dimensions.

### 3.2. Selection of the Optimal MRI Sequence

The same registered coronal plane for the four different MRI acquisitions and µCT with the corresponding ROIs are shown in the first row of Figure 6. The binarized voxels of the respective ROIs are shown in the second row of Figure 6. In Table 3, the morphological parameters derived as “mean ± SD” are reported for each sample, as well as imaging modality with the absolute errors calculated on each feature for both the TSE and GRE sequences performed with respect to the µCT reference. Clearly, TSE sequences provided a better contrast, so the inner trabecular network was more easily identifiable (Figure 6) and this was true for all the scanned samples acquired using both 3T and 7T MRI. Moreover, considering the morphological features quantified on UHF images conducted on the three samples, the corresponding errors, taking μCT as a reference, were lower for the TSE sequence (maximum errors always lower than 12% for all the characteristics) than GRE (maximum errors up to 63% for BVF, 107% for Tb.Th and 35% for Tb.Sp) (Table 3 and Figure 7).

The statistical analysis, conducted on three 2D images per bone sample using the one-way ANOVA, showed no significant statistical difference (*p* > 0.01) between 7T TSE and the μCT reference for the whole set of parameters. The coefficient of determination (R^2^), calculated to address their functional relationship, ranged from 0.52 for Tb.N to 0.81 for Tb.Int. The corresponding bias (considering μCT as a reference) was determined thanks to the Bland–Altman analysis and resulted in a mean bias of 4.8% for the whole set of morphological parameters. The interclass correlation coefficient came as an additional support and all parameters were classified as good (ICC ranging from 0.53 for Tb.Th to 0.73 for Tb.OrP) or excellent (ICC = 0.80 for Tb.Int). Different results were obtained using the 7T GRE images. The one-way ANOVA showed a significant statistical difference for three out of seven morphological parameters analyzed (BVF, Tb.Th and Tb.Int). The coefficient of determination (R^2^) ranged from 0.11 for Tb.Th to 0.63 for Tb.OrP and the Bland–Altman analysis showed higher biases for all the parameters (ranging from 2.0% for Tb.OrS to 51.1% for Tb.Th), with a mean bias of 20.6%. The ICC values were generally lower than those derived using 7T TSE and were classified as poor (BVF, Tb.Th and TbOrS) or good (Tb.Sp, Tb.N, Tb.OrP and Tb.Int), with Tb.Th performing the worst (ICC = 0.11) and Tb.Int the best (ICC = 0.73).

Interestingly, the analysis conducted on a single femur (S1) using both 3T and 7T MRI showed that the corresponding error committed on morphological features quantified on UHF images, taking µCT as a reference, was the lowest, regardless of the magnetic field strength (Figure 5). As an example, BVF error was reduced by 50% and 71% from 3T to 7T for TSE and GRE, respectively. Similar results were obtained for all the conventional morphological parameters, with a mean reduction in the committed error of 58% from 3T to 7T TSE and of 69% from 3T to 7T GRE. The largest reduction was 100% for the Tb.Sp from 3T to 7T TSE and 178% for the Tb.Th from 3T to 7T GRE. The morphological characteristics derived from gray level intensities showed very similar results between 3T and 7T, with errors compared to the µCT reference always lower than 11% for both TSE and GRE. Considering the MRI pulse sequences at 3T, TSE did not systematically perform better than GRE. In fact, errors were reduced for BVF (81%) and Tb.Th (249%) and were increased for Tb.Sp (80%) and Tb.N (17%). Moreover, the results also showed that images acquired at 7T did not always provide better results, as compared to 3T. In fact, when comparing 3T TSE with 7T GRE, the committed errors were lower for BVF (mean error equal to 48% for 3T TSE and 63% for 7T GRE) and for Tb.Th (mean error equal to 33% for 3T TSE and to 107% for 7T GRE).

### 3.3. Correlation between DXA-BMD and Microarchitecture

The linear regression between the morphological parameters and BMD derived from clinical DXA, for the three analyzed samples, was also assessed to determine whether these two analyses provided similar information and, if appliable, to determine the relation with a specific MR sequence (Figure 8).

No linear correlation was assessed between µCT morphological parameters and BMD for the majority of the morphological parameters analyzed (R^2^ equal to 0.20 for Tb.Th, 0.42 for Tb.Sp, 0.01 for Tb.N, 0.16 for Tb.OrP, 0.15 for Tb.OrS). However, BVF and Tb.Int showed, respectively, modest (R^2^ = 0.50) and good (R^2^ = 0.73) correlations. Images from 7T MR showed similar results, i.e., no linear correlation was found for all morphological characteristics using the TSE sequence (R^2^ equal to 0.40 for BVF, 0.23 for Tb.Th, 0.31 for Tb.Sp, 0.16 for Tb.N, 0.01 for Tb.OrP and 0.30 for Tb.OrS) or GRE (R^2^ equal to 0.22 for BVF, 0.02 for Tb.Th, 0.47 for Tb.Sp, 0.01 for Tb.N, 0.38 for Tb.OrP and 0.04 for Tb.OrS) except for trabecular interconnectivity, which presented the highest correlation (R^2^ equal to 0.77 for 7T TSE and 0.66 for 7T GRE).

### 3.4. Correlation between DXA-BMD and Both μCT- and MR- Derived BMD

The BMD derived from µCT images (using both the 3D and 2D approach) and 7T MR images (2D approach only) are presented in Figure 9. The results showed values in the same range of the BMD determined using DXA scans for all the imaging techniques with mean errors equal to 6% for µCT 3D, 15% for µCT 2D, 12% for TSE 2D and 49% for GRE 2D. Moreover, the linear regression between BMD calculated using DXA and the BMD derived from µCT images, using both the 3D and the 2D approach, showed good correlation (R^2^ = 0.94 for 3D µCT and R^2^ = 0.85 for 2D µCT). Similar results were obtained using the 2D approach on 7T TSE images (R^2^ = 0.77), while the BMD derived using 7T GRE showed poor correlation (R^2^ = 0.24).

### 3.5. Correlation between Failure Load and Bone Morphology

We observed one femoral neck and two intertrochanteric fractures with a mean failure load equal to 1733.6 N (SD, 524.6 N) over a mean femoral neck surface of 800.9 mm^2^ (SD, 195.8 mm^2^) derived from high-resolution µCT images in correspondence with the bone fracture site. The resulting mean fracture strain was 2.15 MPa (SD, 0.22 MPa) with a mean work equal to 0.067 J (SD, 0.04 J).

The linear regression between DXA-derived BMD at the femoral neck region and fracture strain showed a poor correlation (R^2^ = 0.43). Higher correlations were found using µCT-derived BMD (R^2^ = 0.66 for the 3D µCT and R^2^ = 0.88 for 2D µCT). A similar result was obtained using the BMD derived from the 2D approach on TSE images (R^2^ = 0.87), while poor correlation (R^2^ = 0.11) was found using GRE images. Moreover, the linear regression between failure strain and microarchitecture parameters derived in the femur head showed excellent to good correlations for both Tb.Sp (R^2^ = 0.97, 0.88 and 0.70, respectively, derived from µCT, 7T TSE and 7T GRE) and Tb.Int (R^2^ = 0.999, 0.792 and 0.934, respectively, for µCT, 7T TSE and 7T GRE) (Figure 10). However, poor correlations were found for BVF (R^2^ < 0.22), Tb.Th (R^2^ < 0.25) and Tb.N (R^2^ < 0.36) among all the used imaging modalities in this study.

## 4. Discussion

The investigation of complete cadaveric bone segments is of interest for a reliable assessment of bone quality and bone disorders. Up to now, cadaveric bone imaging using MRI has been mainly performed in small specimens (<5 cm^3^) [15,40], with the consequence of a poor representative characterization of the entire bone. Investigations of large bone specimens could provide a more complete picture as long as magnetic susceptibility artefact issues due to bubbles trapped inside the trabecular network are removed as previously described [21]. High and ultra-high field MRI is of interest given the high-image resolution that can be achieved using a reasonable acquisition time, but power deposition can be an issue for a certain type of MRI sequence. A comparative analysis between different MRI sequences (performed at different main field strengths) and μCT of proximal femurs’ trabecular network could be of interest for the clinical application of in vivo non-invasive microarchitecture analysis of one of the most invalidating osteoporotic sites, and more in general of deep bone segments.

In the present study, we addressed the issue related to MRI sequence selection at high and ultra-high field for the histomorphometric assessment of large bone segments in vivo.

Considering that previous studies have indicated that TSE sequences were less susceptible to partial volume effects as compared to GRE sequences [13], one might expect different effects for the bone micro-architecture quantification. Our results regarding bone micro-architecture further confirm and extend those from previous studies conducted at 1.5T [41], 3T [13,42] and 7T [11]. The TSE sequence was indeed less prone to partial volume effects errors as compared to GRE and so regardless the magnetic field strength and the investigated bone segment. These stronger partial volume effects on GRE sequences created opposite effects on Tb.Th and Tb.Sp. In fact, Tb.Th computed from GRE images was largely overestimated (249% at 3T and 107% at 7T) with respect to μCT reference values, likely as a result of more trabeculae thickening and the aggregation of the thinnest trabeculae. The same phenomenon, i.e., trabecular thickening and aggregation, led to a space reduction between the biggest trabeculae and also between the thinnest, so that the computed Tb.Sp values were similar to those computed from μCT images. Previous measurements performed at 3T on tibiae and radii indicated that trabeculae are more accurately depicted using the spin-echo-type sequence, whereas gradient echo sequences lead to trabecular broadening [13]. Krug et al. also suggested that trabecular structure can be enhanced using gradient echo sequences so that smaller trabeculae disappear due to partial volume effects [13]. In agreement with the present results, Majumdar et al. [11] also indicated that structural parameters computed from distal tibia images recorded at 7T were closer to reference μCT values as compared to those acquired at 3T. They also reported that spin echo sequences revealed a more homogeneous bone marrow signal than gradient echo ones, hence reducing the susceptibility-induced broadening effects of the trabeculae [11]. Furthermore, they showed that since trabecular structure is overemphasized at a higher field strength due to susceptibility-induced broadening, smaller trabeculae, usually not visible due to partial volume effects, may be emphasized at 7T [11]. Therefore, the application of a higher main magnetic field strength may provide a more accurate trabecular network identification.

Of interest, the morphological parameters computed from TSE images recorded at 7T were in agreement with the μCT reference values for all three acquired femur specimens and with those from previous works [11,12,17,18,19,41,43]. For the whole set of microarchitecture features, the committed errors were always lower than 12% and the corresponding mean bias was low (<5%). In young healthy subjects, Majumdar et al. [11] showed similar tibiae morphological parameters for both gradient and spin echo sequences recorded at ultra-high field (0.45 for BVF, 0.26 mm for Tb.Th and 0.33 mm for Tb.Sp with spin echo, and 0.50 for BVF, 0.33 for Tb.Th and 0.33 for Tb.Sp with gradient echo). Similarly to our results, they showed an overestimation of BVF and Tb.Th for gradient echo as compared to spin echo, while the identification of the Tb.Sp was consistent between the two sequences. Moreover, the values’ differences (higher BVF and lower Tb.Sp) might be due to the difference in conditions and anatomical site (healthy, young, and in vivo tibiae vs. old and cadaveric proximal femurs in our case). Our results are in agreement with those from Tjong et al. [18], in a study conducted using μCT (voxel size 18 μm isovolumetric) and HRpQCT (voxel sizes 41 μm isovolumetric) on cadaveric radii. The corresponding morphological parameters (0.21 ± 0.06 for BVF, 0.75 ± 0.15 mm for Tb.Sp and 0.21 ± 0.01 mm for Tb.Th) were in the same range as those we computed in a different anatomical site.

The linear regression between morphological parameters and BMD assessed for all the different imaging techniques (μCT, 7T TSE and 7T GRE) showed a poor correlation (R^2^ < 0.47 for all the parameters except Tb.Int (R^2^ between 0.66 and 0.77 for the three different imaging modalities)). Among the conventional morphological parameters, higher correlations were found for Tb.Sp, in particular for 7T GRE images (R^2^ equal to 0.47), while Tb.Th usually showed the lowest. Similar effects have been assessed previously. Majumdar et al. [12] showed moderate correlations for Tb.Sp (0.41, respectively) on in vivo distal radii acquired using a modified gradient echo imaging sequence at 1.5T. Moreover, Chang et al. showed no linear regression between microarchitectural parameters and BMD on a study conducted on the hip of 60 postmenopausal woman scanned using 3T MRI [1]. The poor correlation assessed between the morphological parameters and the BMD extended previous results showing that bone density and structure are two distinct characteristics. However, both are essential for defining the clinical state of bone [1,9,12]. The highest correlations were found for trabecular interconnectivity, which provided good correlation for all the three different imaging modalities. The progressive decrease in trabecular interconnectivity with lower BMD may suggest a reduced trabecular orientation profile and a reduced dynamic spread of impulsive actions. On that basis, Tb.Int seemed to suggest an increased bone fragility and a consequent increased risk of fracture.

The 2D approach for deriving the BMD was proposed due to the fact that MR images are usually acquired using a slice thickness at least 3 times bigger than the pixel size, a distance factor/slice gap equal to the slice thickness to reduce the cross-talk effects, and a single acquisition using 7T TSE is not able to depict the whole femur head segment. Therefore, the potential to derive a BMD approximation using the central coronal plane only of the proximal femur is of great interest. The correlation between BMD derived from DXA scans and computed using the BMC from 2D and 3D µCT, and 2D 7T MR images, in approximately the same femoral neck ROI, showed good to moderate correlation for all the techniques but 7T GRE. In particular, the BMDs computed from 2D TSE images showed values in the same range of conventional DXA with a comparable standard deviation. Similar results have been assessed previously by Kroger et al., in a study conducted in 32 volunteers comparing BMD derived from DXA and 1.5T MR images in the lumbar vertebra L3, assessing a good correlation (r = 0.665) [34]. The results were further confirmed and extended by Arokoski et al. in a study conducted in the femoral neck of 28 people using DXA and a spin echo sequence (TR/TE = 730/11 ms) at 1.5T MRI, where BMD derived from these two techniques provided a good correlation (r = 0.74) [33]. The results seemed to suggest that MR images can be used to assess both bone microstructure and mineral density in multiple bone regions with good accuracy compared to gold-standard X-ray techniques.

The mechanical tests provided failure load values in the same range as those from previous studies [9,37,39]. Moreover, the linear regression between failure load and BMD derived using DXA was similar to a previous study conducted by Guenoun et al. on 10 samples (R^2^ = 0.415) [9]. Interestingly, the linear regression between the fracture strain and both trabecular spacing and trabecular interconnectivity showed good correlation, regardless of the imaging modality (R^2^ > 0.79). This result clearly supports the hypothesis that microarchitectural parameters could provide additional information to BMD on the bone health and fracture risk assessment.

Our results showed that the registration between MRI and µCT images converged for both GRE and TSE images recorded at both high and ultra-high fields. Previous studies related to multimodal registration have been conducted for images with similar voxel dimensions and used approaches based on entropy, correlation, and pixels’ intensity differences [44,45]. More recently, Wafi et al. reported that NCC was a robust registration approach based on illumination changes. However, the corresponding computational cost was high, and the technique was sensitive to thin-line structures, as reported by Penney et al. [25,45,46]. Our results related to the registration process showed that TSE images always provided a better result, for both NCC and ΔIm, as compared to GRE images, regardless of the magnetic field strength. This better performance might also be related to voxel resolution for both TSE vs. GRE and 3T vs. 7T. In fact, for a given field strength, a reduced in-plane resolution (0.13 vs. 0.18 mm at 7T and 0.21 vs. 0.23 mm, respectively, for TSE and GRE at 7T and 3T) manifests in greater NCC scores and lower ΔIm standard deviations. However, while an increased NCC score and ΔIm closest to the reference were obtained for GRE images from 3T to 7T, an opposite effect was computed for TSE images, i.e., a slightly decreased NCC and a different mean ΔIm value as compared to the expected values, while the standard deviation decreased from 16 to 7, corresponding to a decreased μCT range of 0.46 mm. This effect might be explained by the slice thickness (from 1.1 mm at 3T to 1.5 mm at 7T), which is due to the smaller number of acquirable images. Zhao et al. supported this hypothesis, showing that images with the same resolution (both in-plane and through-planes) can be registered accurately with more confidence [47]. Therefore, the NCC method can be considered as an optimized approach for images, i.e., μCT and MRI with a 20- to 30-fold difference in slice thickness.

Some limitations must be acknowledged in the present study. Due to the specific absorption rate (SAR), the number of images acquirable using TSE at 7T was strictly limited. In order to acquire the whole bone segment, multiple acquisitions could be performed as long as pauses are allowed between acquisitions. Moreover, although tissue alterations due to the sample freezing–defreezing process could have been expected, our morphological evaluation provided results similar to those in previous studies conducted in vivo and ex vivo [41,42] suggesting a limited effect of this process. Finally, the study was conducted in a limited number of cadaveric femurs. The number was limited for obvious ethical reasons.

## 5. Conclusions

Overall, in agreement with previous studies, the present results obtained from three cadaveric proximal femur heads showed that TSE images were less prone to partial volume effects and trabecular broadening than GRE images, regardless of the magnetic field strength. Our results extended previous results and showed that the higher the field strength, the smaller the committed errors and the larger the agreement with μCT reference values. Furthermore, the morphological and statistical analyses showed that the TSE sequence at 7T could be used in vivo for evaluating the bone microstructure of deep bone segments, with a very good approximation as compared to the μCT gold standard. Moreover, our results showed that the BMD derived from both µCT and 7T TSE images positively correlated with the clinical standard BMD derived using DXA and that trabecular interconnectivity presented higher correlation to fracture load than those derived using BMD.

Therefore, our results clearly suggest that UHF MRI could be of interest as an in vivo and non-invasive imaging modality for the assessment of both bone microarchitecture and mineral density and could provide a more comprehensive view of bone quality and fracture risk than DXA alone.

## Figures and Tables

**Figure 1 diagnostics-11-01603-f001:**
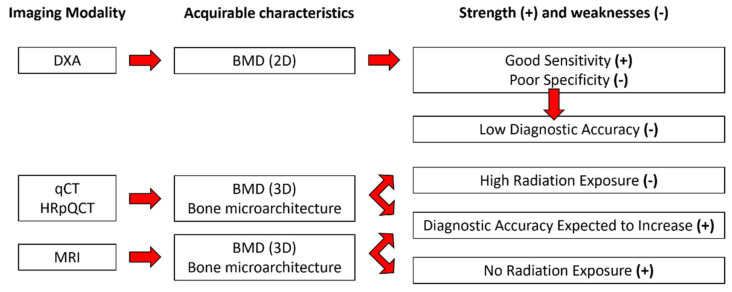
Available characteristics from each imaging modality with their corresponding strength (+) and weaknesses (-).

**Figure 2 diagnostics-11-01603-f002:**
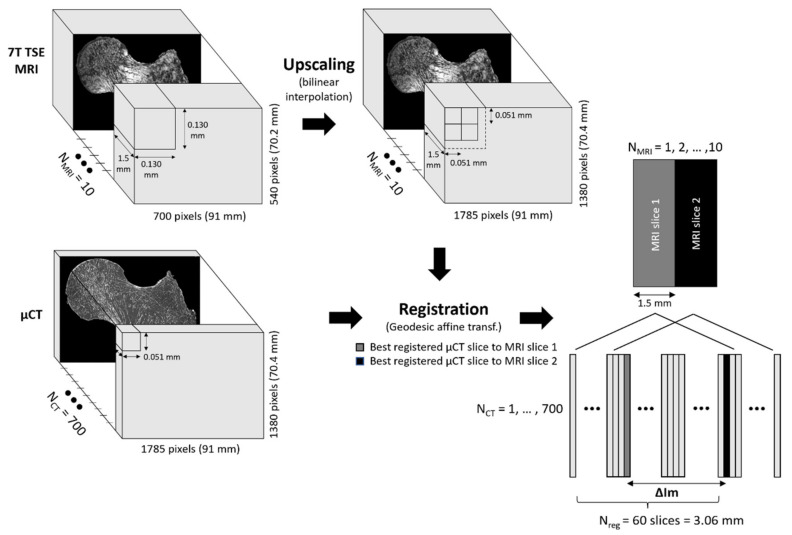
Registration workflow.

**Figure 3 diagnostics-11-01603-f003:**
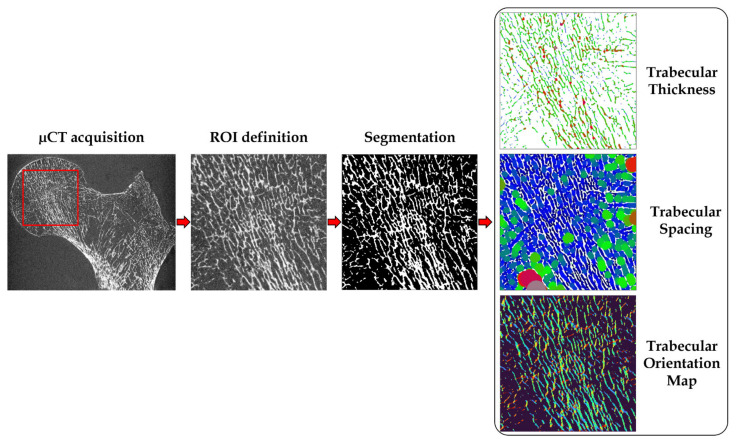
Preprocessing and elaboration steps for the microarchitecture characteristics assessment.

**Figure 4 diagnostics-11-01603-f004:**
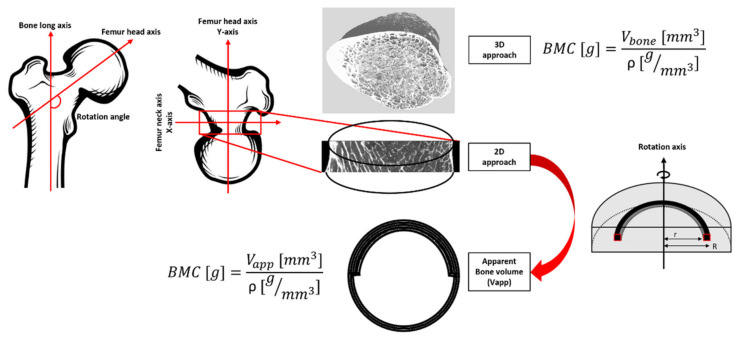
Workflow for bone mineral density estimation.

**Figure 5 diagnostics-11-01603-f005:**
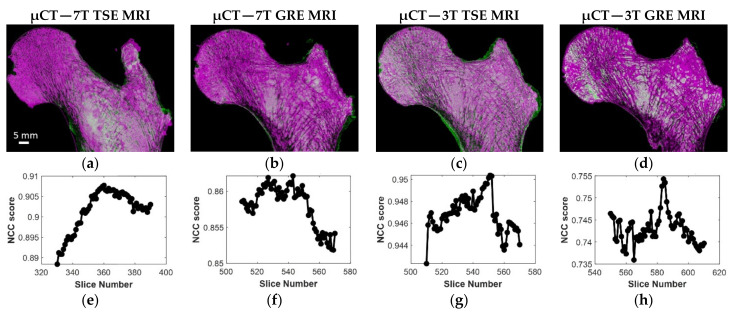
(**a**–**d**) S1 best registration for the four different MRI acquisitions with μCT; (**e**–**h**) corresponding NCC efficiency profile.

**Figure 6 diagnostics-11-01603-f006:**
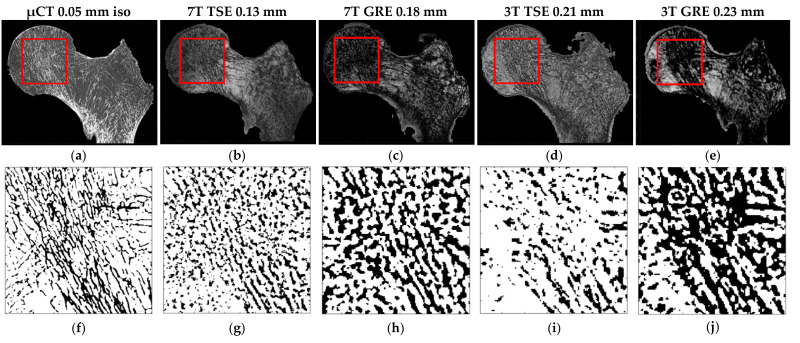
(**a**–**e**) S1 same coronal planes of the (**a**) μCT and four different MRI acquisitions ((**b**) 7T TSE, (**c**) 7T GRE, (**d**) 3T TSE and (**e**) 3T GRE). The red square identify the ROI extrapolated from all the registered images. (**f**–**j**) Corresponding ROI binarized (automatic local thresholding with a window size of 10 × 10 pixels).

**Figure 7 diagnostics-11-01603-f007:**
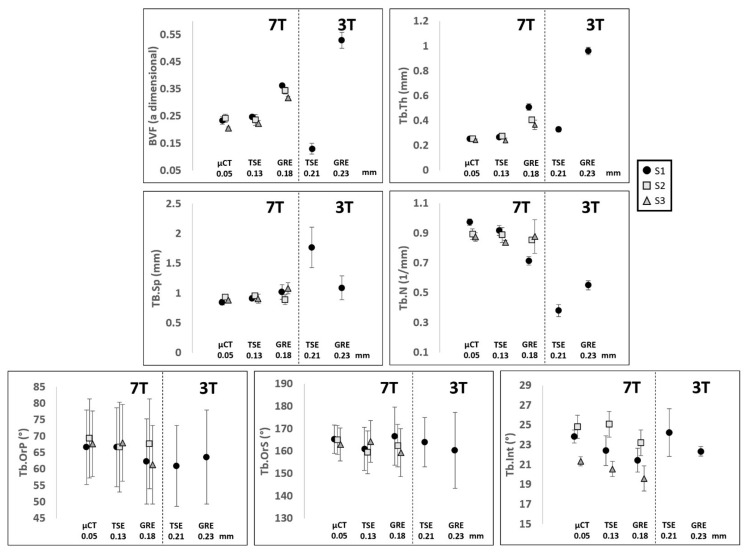
Data shows mean and SD of, respectively, BVF: bone volume fraction, Tb.Th: trabecular thickness, Tb.Sp: trabecular spacing, Tb.N: trabecular number, Tb.OrP: principal trabecular orientation, Tb.OrS: secondary trabecular orientation and Tb.Int: trabecular interconnectivity of the three analyzed samples S1 ‘●’, S2 ‘■’ and S3 ‘▲’ scanned with different scanners (µCT and MRI), field strengths (3T and 7T), sequences (TSE and GRE) and resolution.

**Figure 8 diagnostics-11-01603-f008:**
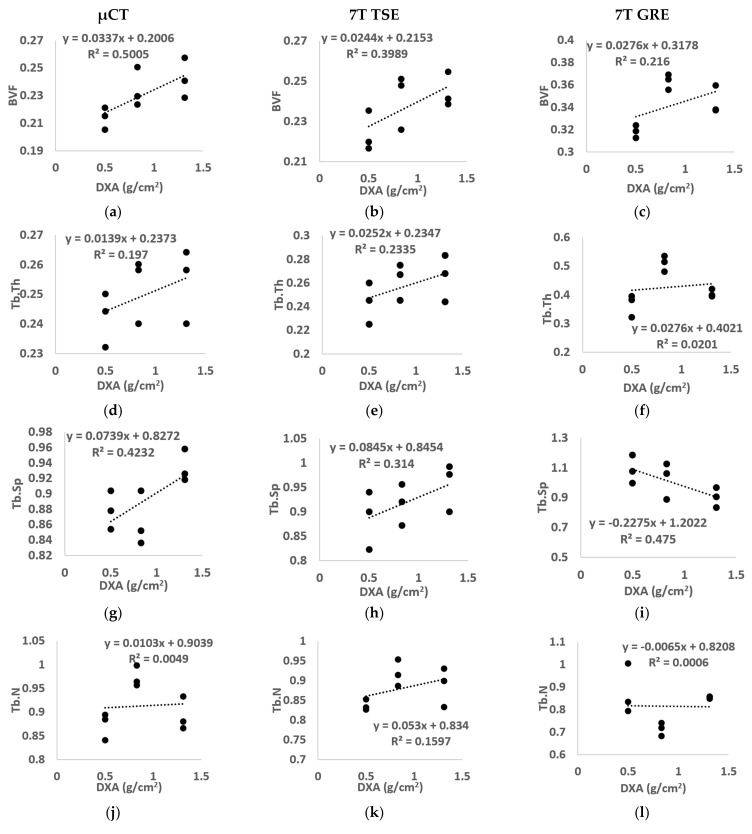
Linear regression between BMD calculated using standard DXA and all morphological parameters ((**a**–**c**) BVF, (**d**–**f**) Tb.Th, (**g**–**i**) Tb.Sp, (**j**–**l**) Tb.N, (**m**–**o**) Tb.OrP, (**p**–**r**) Tb.OrP, (**s**–**u**) Tb.Int) derived from µCT (left), 7T TSE MR (middle) and 7T GRE MR (right) images. “BVF” refers to bone volume fraction, “Tb.Th” refers to trabecular thickness, “Tb.Sp” refers to trabecular spacing, “Tb.N” refers to trabecular number, “Tb.OrP” refers to principal trabecula orientation, “Tb.OrS” refers to secondary trabecular orientation, “Tb.Int” refers to trabecular interconnectivity.

**Figure 9 diagnostics-11-01603-f009:**
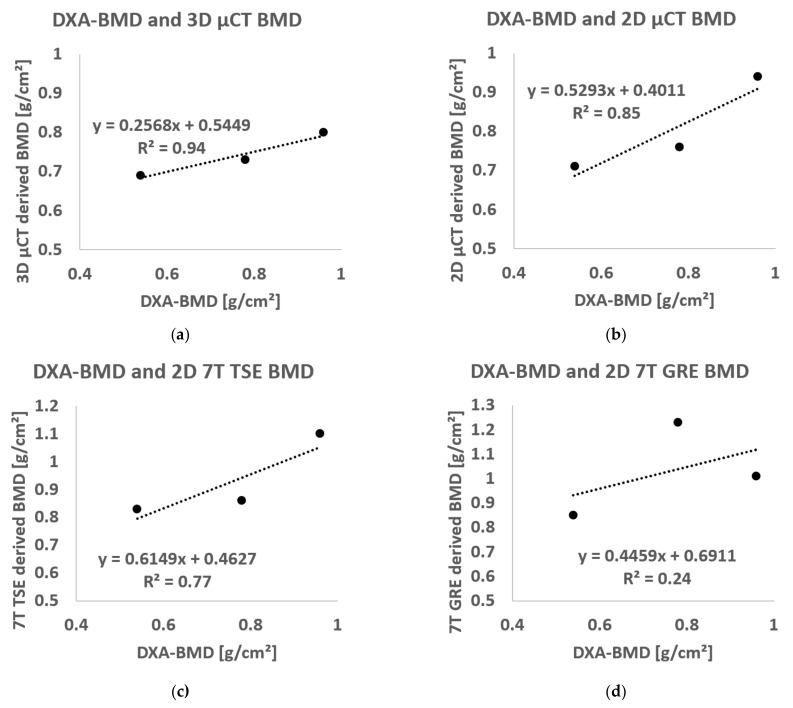
Linear regression between BMD calculated using standard DXA analysis and BMD derived using (**a**) µCT 3D approach, (**b**) µCT 2D approach, (**c**) TSE 2D approach and (**d**) GRE 2D approach.

**Figure 10 diagnostics-11-01603-f010:**
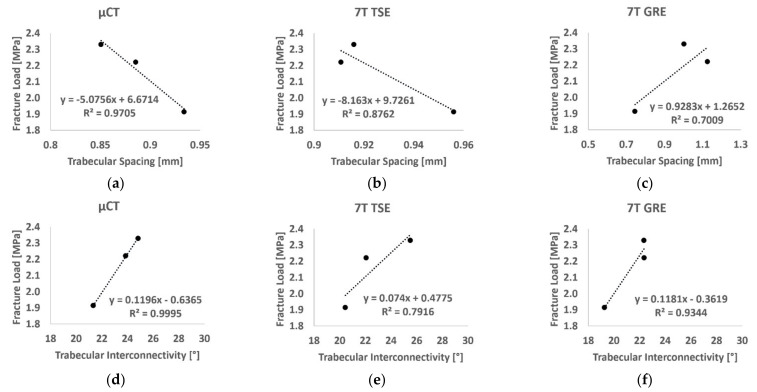
Linear regression between fracture load and trabecular spacing (**a**–**c**) and trabecular interconnectivity (**d**–**f**) derived from µCT images (left), 7T TSE (middle) and 7T GRE (right).

**Table 1 diagnostics-11-01603-t001:** List of main parameters used for MRI acquisitions.

Seq.	TR/TE (ms)	Flip Angle (°)	FoV (mm)	Bandwidth (Hz/Px)	NeX	Voxel Size (mm)	Slice Thickness (mm)	Slices	Acq.Time (min:sec)
7T TSE	1040/14	150	97 × 130	244	2	0.13 × 0.13	1.5	10	17:45
7T GRE	11/5.60	12	120 × 175	330	3	0.18 × 0.18	1	48	9:27
3T TSE	1170/12	140	119 × 119	255	2	0.21 × 0.21	1.1	36	16:45
3T GRE	16.5/7.78	10	120 × 120	130	2	0.23 × 0.23	1.1	40	11:17

TR: repetition time, TE: echo time, FoV: field of view and NeX: number of excitations.

**Table 2 diagnostics-11-01603-t002:** S1 normalized cross-correlation (NCC) scores and ΔIm, the number between two consecutive best registered μCT images used to evaluate the registration efficiency for the four stacks of MRI images.

S1	7T TSE—μCT	7T GRE—μCT	3T TSE—μCT	3T GRE—μCT
**NCC score**	0.93 ± 0.01	0.86 ± 0.01	0.94 ± 0.01	0.75 ± 0.01
**ΔIm**	67 ± 7 (59)	36 ± 13 (39)	42 ± 16 (43)	50 ± 18 (43)

ΔIm are presented as mean ± SD. expΔIm values are presented in brackets.

**Table 3 diagnostics-11-01603-t003:** Morphological characteristics between registered µCT—7T MR images.

		BVF	Tb.Th	Tb.Sp	Tb.N	Tb.OrP	Tb.OrS	Tb.Int
**S1 (BMD-DXA = 0.83 g/cm^2^)**	µCT	0.23 ± 0.01	0.25 ± 0.01	0.86 ± 0.04	0.97 ± 0.03	67 ± 11	165 ± 6	23.84 ± 0.66
	7T TSE	0.24 ± 0.01	0.26 ± 0.02	0.92 ± 0.04	0.92 ± 0.03	67 ± 13	161 ± 10	22.41 ± 1.50
	Max diff%	8%	6%	8%	8%	1%	3%	9%
	7T GRE	0.36 ± 0.01	0.50 ± 0.03	1.02 ± 0.03	0.71 ± 0.03	62 ± 12	167 ± 10	21.44 ± 1.20
	Max diff%	63%	107%	27%	32%	10%	5%	13%
**S2 (BMD-DXA = 1.31 g/cm^2^)**	µCT	0.24 ± 0.01	0.25 ± 0.01	0.93 ± 0.02	0.89 ± 0.04	69 ± 12	165 ± 7	24.83 ± 1.18
	7T TSE	0.24 ± 0.02	0.26 ± 0.01	0.96 ± 0.05	0.89 ± 0.05	67 ± 14	160 ± 9	25.08 ± 1.31
	Max diff%	10%	7%	5%	7%	7%	4%	5%
	7T GRE	0.34 ± 0.01	0.40 ± 0.02	0.89 ± 0.08	0.85 ± 0.03	68 ± 14	162 ± 10	23.21 ± 1.29
	Max diff%	57%	64%	13%	9%	4%	3%	8%
**S3 (BMD-DXA = 0.50 g/cm^2^)**	µCT	0.21 ± 0.01	0.24 ± 0.01	0.88 ± 0.03	0.87 ± 0.03	68 ± 10	163 ± 7	21.33 ± 0.46
	7T TSE	0.22 ± 0.01	0.24 ± 0.02	0.89 ± 0.06	0.84 ± 0.02	68 ± 12	164 ± 11	20.56 ± 0.77
	Max diff%	11%	12%	5%	6%	3%	2%	6%
	7T GRE	0.32 ± 0.01	0.37 ± 0.04	1.08 ± 0.09	0.88 ± 0.11	61 ± 12	159 ± 9	19.60 ± 1.27
	Max diff%	55%	70%	35%	12%	17%	9%	12%

Morphological characteristics are expressed as mean ± SD for the three registered µCT-7T MRI images for all the three different samples (S) for both turbo spin echo (TSE) and gradient recalled echo (GRE) with the corresponding maximum percentage difference (max diff%). BVF: bone volume fraction, Tb.Th: trabecular thickness, Tb.Sp: trabecular spacing, Tb.N: trabecular number, Tb.OrP: principal trabecular orientation, Tb.OrS: secondary trabecular orientation, Tb.Int: trabecular interconnectivity.

## Data Availability

The data presented in this study are available on request from the corresponding author. The data are not publicly available due to restrictions imposed by Aix Marseille University and by the local ethics committee regarding patients data sharing.

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
