# Peer review of "Validation and Optimization of Proximal Femurs Microstructure Analysis Using High Field and Ultra-High Field MRI"

_diagnostics, 2021, doi:10.3390/diagnostics11091603_

Round 1
Reviewer 1 Report
The manuscript “Validation and Optimization of Proximal Femurs Microstructure Analysis using High Field and Ultra-High Field MRI” presents interesting research on bone microstructure and bone mineral density using turbo spin echo and gradient recalled echo MRI sequences. The results are exciting for MRI imaging of bone microstructures. The manuscript is well written and looks scientifically sound, therefore recommended for acceptance.
Line 19, Abbreviation used first time should be expanded e.g., DXA
Reviewer 2 Report
General comment:
This work deals with the comparative analysis of bone microstructure parameters and bone mineral density (BMD), computed using MRI, with different approaches and field strengths, and computed with μCT and DXA. An 8% error for trabecular thickness with no significative statistical
difference and good intraclass correlation coefficient, with μCT, was found for the parameters. The results showed no correlation with DXA-related parameters. The study proposes MRI as a diagnostic tool for bone microstructure assessment.
Specific comment throghout the paper:
Abstract
Line 19: DXA acronym was not defined
1. Introduction
Lines 41-55: A summary table for these findings could be useful to the reader.
Lines 61-62: Provide some details of why these sequences are used, i.e., highlight the physical rationale behind their use for MRI bone imaging.
Line 67: The symbol "r" for the correlation coefficient has to be defined.
Lines 72-75: Missing ref. for the power deposition issues of UHF MRI. I suggest the author to ride and cite:
10.23919/ROPACES.2017.7916404
P.M. Robitaille and L. Berliner, Ultra High Field Magnetic Resonance Imaging, 2006.
Line 82: There is a gap at this point of the Introduction. The "aim and scope" sub-section should be better introduced, in my opinion, for the sake of clarity and to improve the readability.
2. Materials and Methods
2.1. Sample Preparation
Line 98: Please correct the superscripts for the BMD unit of measure.
2.2. Imaging
2.2.1. μCT Imaging
Ok to me.
2.2.2. MRI Imaging
Line 125: Please provide the sequence parameters in an explicit way. You can use the appendix.
Line 127: Please report the voxel size value.
Tab. 1: TR, TE, NeX not defined anywhere.
2.3. Image Analysis
2.3.1. Image Registration
Line 132: About co-registering, please better explain the moethod used. I suggest the author to read and cite
https://doi.org/10.3390/app9153156
They can use the same way of presenting the co-registration operations.
Line 143: The version of Matlab is not reported. Please modify.
Line 147: The symbol delta Im is not defined. Please fix.
2.3.2. Bone Morphological Quantification
LINE 161: What about the automatic local thresholding technique? Please provide additional details for you problem.
Line 171: Missing details for iMorph (e.g., version, etc.).
Line 180: Missing coma - "assessed, i.e., ..."
Line 191: "build in" - hange in "built in". If you don't provide the code these info are quite useless and do not esnure reporoducibility. I suggest the author to try to explain the curve fitting procedure, while also discussing why they are not using the avialable fitting tools.
A figure which summarizes and elucidates the steps described in Sect. 2.3.2 can be of help to the reader, while increasing the quality of your work.
2.3.3. Bone Mineral Density Assessment
Line 246: Use the same notation of Fig. 2 and use the subscript.
3. Results
3.1. Registration Quality
Line 283: The number in bracket parentheses is not clear to me. What does it represent?
3.2. Selection of the Optimal MRI Sequence
Fig. 4, Line 343: This is what miss when you introduced the automatic local thresholding alg. The definition of the functioning and the paramters such as window size, etc., are relevant to understand the quality of your results.
4. Discussion
Lines 422 , 426 and 454, 472, 492, 520, 544: Extra spaces. Please fix.
Round 2
Reviewer 2 Report
The authors replied to all my questions, doubts and comments in an exhaustive way, performed a thorough review of the manuscript and modified it to improve the quality and value of the work.